# Transcriptome Analysis of the Influence of High-Pressure Carbon Dioxide on *Saccharomyces cerevisiae* under Sub-Lethal Condition

**DOI:** 10.3390/jof8101011

**Published:** 2022-09-27

**Authors:** Tonghuan Yu, Ukyo Takahashi, Hitoshi Iwahashi

**Affiliations:** 1Key Laboratory of Yellow River and Huai River Water Environment and Pollution Control, Ministry of Education, Henan Key Laboratory for Environmental Pollution Control, International Joint Laboratory on Key Techniques in Water Treatment, School of Environment, Henan Normal University, Xinxiang 453007, China; 2The United Graduate School of Agricultural Science, Gifu University, Yanagido 1-1, Gifu 501-1193, Japan; 3Faculty of Applied Biological Science, Gifu University, Yanagido 1-1, Gifu 501-1193, Japan

**Keywords:** high-pressure carbon dioxide, *Saccharomyces cerevisiae*, cell membrane, transcriptome analysis, RNA sequencing, ontology analysis

## Abstract

High-pressure carbon dioxide (HPCD), a novel non-thermal pasteurization technology, has attracted the attention of scientists due to its high pasteurization efficiency at a lower temperature and pressure. However, the inactivation mechanism has not been well researched, and this has hindered its commercial application. In this work, we used a sub-lethal HPCD condition (4.0 MPa, 30 °C) and a recovery condition (30 °C) to repair the damaged cells. Transcriptome analysis was performed by using RNA sequencing and gene ontology analysis to investigate the detailed lethal mechanism caused by HPCD treatment. RT-qPCR analysis was conducted for certain upregulated genes, and the influence of HPCD on protoplasts and single-gene deletion strains was investigated. Six major categories of upregulated genes were identified, including genes associated with the pentose phosphate pathway (oxidative phase), cell wall organization or biogenesis, glutathione metabolism, protein refolding, phosphatidylcholine biosynthesis, and AdoMet synthesis, which are all considered to be associated with cell death induced by HPCD. The inactivation or structure alteration of YNL194Cp in the organelle membrane is considered the critical reason for cell death. We believe this work contributes to elucidating the cell-death mechanism and providing a direction for further research on non-thermal HPCD sterilization technology.

## 1. Introduction

Currently, consumers’ requirements for food nutritional value and organoleptic quality are increasing. Traditional pasteurization technology involves the use of high temperatures, which can lead to a decrease in food nutritional and organoleptic quality [1]. Furthermore, the high-temperature application is generally associated with high energy consumption, which is counterproductive in terms of the United Nations Sustainable Development Goals [2]. Due to these limitations, an increasing number of novel pasteurization technologies, such as high hydrostatic pressure, pulsed light technology, microwave radiation, pulsed electric fields, ultrasound, UVC light-emitting diodes, and high-pressure carbon dioxide technology, have attracted the attention of scientists and researchers [3,4,5,6,7,8,9].

In the field of non-thermal decontamination technologies, high hydrostatic pressure is the most developed and widely used technology in the food industry. The damage to cell membranes due to high-hydrostatic-pressure treatment has been considered the main reason for microbial inactivation [10,11]. However, due to the high hydrostatic pressure required (100–600 MPa) when applying this technology to commercial pasteurization, the increased requirements for equipment become a limitation. In addition, the application of high pressure makes it difficult to operate on a large commercial scale. Therefore, high-pressure carbon dioxide technology has gained a significant amount of attention due to several advantages, such as lower pressure application (less than 20 MPa), lower equipment requirement, and lower temperature application (less than 50 °C); additionally, carbon dioxide is relatively inexpensive, it is a non-toxic system, and it has a lower impact on nutrients [1]. 

However, compared to high hydrostatic pressure, HPCD pasteurization has yet to reach a large commercial scale, due to a lack of substantiating research. Until recently, it was assumed that cell-membrane damage is the leading lethal cause of HPCD pasteurization; however, the detailed mechanism remains unknown. According to the research conducted by the authors of Reference [12] on high hydrostatic pressures, yeast cells present different responses to pressure under growth inhibition and lethal conditions. Therefore, we suggest that the yeast cells may present different responses to different HPCD conditions. The different HPCD treatment conditions for yeast cells were divided into growth-inhibition conditions, lethal conditions, and sub-lethal conditions. The HPCD growth-inhibition condition is defined as the condition in which the yeast cells’ growth rate began to decrease significantly after HPCD treatment compared to the cells experiencing no HPCD treatment. The conditions of 0.5 MPa at 25 °C for 2 h can result in growth inhibition [8]. The HPCD lethal condition is defined as conditions that can induce entire cell death, and subsequent cell recovery on proper culture medium after treatment was impossible. This condition may result in the cell losing the ability to synthesize new RNA. In contrast, the sub-lethal HPCD condition does not induce entire cell death, and the cells can recover from the damage when placed in a proper culturing environment [13,14]. Sub-lethal damage of cells is considered an important point for investigating the cell’s various physical changes and mechanisms associated with cell death [15,16]. Our previous study proved that 4.0 MPa at 30 °C for 4 h are the sub-lethal HPCD conditions for yeast cells [17]. Therefore, in this study, yeast cells exposed to sub-lethal conditions for 4 h, followed by 1 h of recovery incubation (30 °C), were selected for transcriptome analysis to understand the mechanisms underlying HPCD-mediated cell death. Furthermore, RT-qPCR analysis was performed for certain upregulated genes, and the influence of HPCD on protoplasts and single-gene deletion strains was investigated to provide more evidence for transcriptome analysis. 

## 2. Materials and Methods

### 2.1. Strains and Culture Media 

All the strains used in the study are presented in Table 1. A wild-type strain of *Saccharomyces cerevisiae BY4741* (Thermo Fisher Scientific K. K, Tokyo, Japan) was used in all experiments, while other single-gene deletion strains (*S.cerevisiae BY4741 Δzwf1, S. cerevisiae BY4741 Δopi3, S. cerevisiae BY4741 Δgpx1, S. cerevisiae BY4741 Δgsh1, S. cerevisiae BY4741 Δhsp104, S. cerevisiae BY4741 Δcrh1, S. cerevisiae BY4741 Δchs5, S. cerevisiae BY4741 Δgas1*, and *S. cerevisiae BY4741 Δkre6*) were used only for spot assays. The open reading frame (ORF) knockouts were produced by the Saccharomyces Genome Deletion Project [18]. A PCR-based strategy was used to replace each ORF with a KanMX cassette containing unique tags (barcodes) for each deletion.

Strains were cultured by using yeast extract–peptone–dextrose (YPD) liquid medium (10 g/L yeast extract, 20 g/L peptone, and 20 g/L glucose). The pre-culture was further diluted (1:50) with fresh YPD medium in a sterile tube sealed with a silicone sponge-tapered plug. Cultivation conditions were 30 °C at 100 rpm until logarithmic phase (OD_600_ = 0.8–1.0).

### 2.2. HPCD Treatment

Cultured strains were transferred to a high-pressure vessel (30-11HF4; High-pressure Equipment, Elie, PA, USA), which was incubated in a 30 °C thermostatic bath for 1 h to 12 h under HPCD at 4.0 MPa. Our former work proved that those are sub-lethal conditions for *S. cerevisiae* strains [17]. After 1, 2, and 4 h of treatment under HPCD at 4.0 MPa, an incubation for recovery was performed for 1 h at 30 °C. 

### 2.3. RNA Extraction and Sequencing 

The total RNA of each sample was extracted by using the FastRNA Pro Red kit (MP Biomedicals, Santa Ana, CA, USA) and purified by using the RNeasy Mini Kit (Qiagen, Hilden, Germany). The RNA products obtained from the samples were quantified and qualified by using an Agilent 2100 Bioanalyzer (Agilent Technologies, Palo Alto, CA, USA), NanoDrop (Thermo FisherScientific Inc.), and 1% agarose gel electrophoresis before next-generation sequencing of library preparations. RNA extraction, quantification, and qualification were each conducted more than three time, followed by mixing for further library preparation. Approximately 1 μg of extracted total RNA was used for library preparation, based on the manufacturer’s protocol of the NEBNext Ultra RNA Library Prep kit for Illumina. The library sequencing was conducted on the Illumina HiSeq X10 sequencing platform. 

### 2.4. Differential Gene Expression and Enrichment Analysis

The Phred quality score (Q score) was used as an index for the base-calling accuracy and calculated by using the FastQC software (v0.10.1), based on Equation (1): Qphred = −10log10(e) (1)

Quality control for the sequencing reads was performed by using Trimmomatic (v0.30), whereby low-quality reads containing adapter primers and poly-N were removed to generate clean data. The reference genome sequences and gene model annotation files of *S. cerevisiae S288C* were downloaded from Ensemble (http://igenomes.illumina.com.s3-website-us-rast-1.amazonaws.com/Saccharomyces_cerevisiae/Ensembl/R64-1-1/Saccharomyces_cerevisiae_Ensembl_R64-1-1.tar.gz accessed on 16 January 2020) for the reads’ mapping. 

Paired-end clean reads were aligned to the reference genome via the Hisat2 software (v2.0.1). RPKM (reads per kilobase per million reads) values were calculated and considered as the indicators of gene expression abundance. The DESeq2 Bioconductor package, a model based on the negative binomial distribution, was used for the differential expression analysis. After normalization, an HPCD-treated/untreated gene with an expression level >2.0 was defined as an upregulated gene, and a gene with an expression <0.50 was defined as a downregulated gene. 

Metascape (https://metascape.org/gp/index.html#/main/step1 accessed on 28 October 2020) was used for identifying gene ontology (GO) terms that annotate a list of enriched genes of −log_10_ (*p*-value) > 2.

The analysis of different upregulated genes’ related pathways was based on the Saccharomyces Genome Database (https://yeastgenome.org/ accessed on 28 October 2020). 

### 2.5. RT-qPCR

Candidates of the upregulated genes associated with the glutathione synthesis pathway, phospholipid synthesis pathway, cell-wall synthesis pathway, pentose phosphate pathway, and protein refolding were selected to validate the transcriptome data by using quantitative real-time PCR (RT-qPCR). The list of the genes and primers is displayed in Table 2. Amplification and detection were performed by using a StepOnePlus™ Real-Time System (Applied Biosystems, Foster City, CA, USA). 

The reaction mixtures consisted of 5 μL of SYBR Premix Ex Taq II (Takara Bio., Kyoto, Japan), 1 μL of each primer (50 μM), 1 μL of prepared cDNA, and 2 µL of nuclease-free water, which made up a total volume of 10 μL. 

Forty cycles were run for the PCR program after primary denaturation at 95 °C for 10 s. The conditions were as follows: 95 °C for 10 s, 55 °C for 10 s, and 72 °C for 10s. The β-actin gene, *ACT1*, was used as an internal gene-expression control. The whole experiment was repeated three times. The gene-expression level of the non-treated group was defined as 1.0, and the expression level in the HPCD-treatment group (4.0 MPa, 4 h) was evaluated.

### 2.6. Spot Assay

We investigated the susceptibility of the parent strain (*BY4741*) to HPCD and compared it to that of the gene-deficient strains that were selected based on the results of the gene expression analysis. After 12 h of treatment under HPCD (4.0 MPa, 30 °C), serially diluted yeast cells (10^0^–10^5^-times dilution) were used for spotting. A total of 10 µL of the diluted samples was spotted on the YPD agar medium (15 g/L agar, 10 g/L yeast extract, 20 g/L peptone, and 20 g/L glucose). Spotted plates were incubated at 30 °C, and colony formation was observed after 48 h.

### 2.7. Protoplast Preparation and Measurement of Viable Cell Count

Protoplast preparation and cultivation were performed as previously reported [19]. Cells in the late log phase were harvested, washed in water, and incubated in 20 mM 2-mercaptoethanol, 10 mM EDTA, and 10 mM Tris-HCl (pH 7.5 at 30 °C for 30 min). After they were washed with 0.6 M KCl, the cells were suspended in protoplast isolation buffer consisting of 0.6 M KCI and 10 mM Tris-HC1, with a pH 7.5. Zymolyase 20T (Nacalai Tesque Inc., Kyoto, Japan) was added at 200 µg/mL, and the cell suspension was incubated at 30 °C, for 1 to 1.5 h, with gentle shaking. Thereafter, protoplasts were washed with the protoplast isolation buffer. 

For the examination of sensitivity to HPCD, 0.1 mL portions of the protoplasts from a particular strain were embedded in the YPD agar (3%, *w*/*v*) medium containing 0.6 M KCl after HPCD treatment (4.0 MPa, 1–4 h). To measure the frequency of protoplast regeneration, 100 µL of the appropriate dilutions of protoplasts and 0.6 M KCl solution were embedded in triplicate in the regeneration agar plates, and after 2 to 3 days of incubation at 30 °C, the developed colonies were counted.

### 2.8. Microscopy Analysis 

Microscopic imaging was performed by using a phase-contrast microscope (BX-53; Upright Microscope, Olympus, Japan). Images were resized to 400 × 300 pixels.

## 3. Results

### 3.1. Differential Gene Expression Analysis

The wild-type yeast strain was treated with 4.0 MPa 30 °C HPCD for 1, 2, and 4 h, followed by RNA extraction and library preparation. Thereafter, RNA sequencing was conducted on the Illumina HiSeq X10 sequencing platform. The RNA sequencing showed that the Phred quality score for 93% of the clean reads in each sample is more than 30, indicating that the base-calling accuracy for each group was 99.9%, which met the requirement for the next analysis (Table 3).

To identify differentially expressed genes (DEGs), transcriptome analysis was performed for the yeast cells after 1, 2, and 4 h of treatment, respectively. As shown in Figure 1a, 1582 DEGs were identified in the sample after 1 h of treatment, 993 DEGs were identified in the sample after 2 h of treatment, and 1850 DEGs were identified in the sample after 4 h of treatment. Furthermore, 500 DEGs were identified in all three comparative groups, and they can be considered to be the key genes associated with cell death resulting from the HPCD treatment (Figure 1a). A comparison of upregulated genes and downregulated genes is shown in Figure 1b. A total of 875 upregulated and 725 downregulated genes were identified in the cells treated for 1 h, 539 upregulated and 454 downregulated genes were identified in the cells treated for 2 h, and 1050 upregulated and 800 downregulated genes were identified in the cells treated for 4 h. The DEGs associated with upregulation were the highest in expression after 4 h of HPCD treatment.

### 3.2. GO Enrichment Analysis of the DEGs

To better understand the functions of the DEGs, gene annotation, including homologous protein annotation, was conducted by using the GO database. The GO analysis of the upregulated genes is shown in Appendix A, and the GO analysis of the downregulated genes is shown in Appendix A. The yeast cells were exposed to sub-lethal HPCD conditions, which induced the upregulation of some key genes to protect the cells from HPCD conditions. Although some genes were downregulated under environmental stress to prevent the lesion from worsening, many of them were correlated with the cell growth rate. Hence, the upregulated genes were mainly analyzed [20].

Upregulated genes with large −log_10_
*p*-values were selected. The major metabolic pathways and terms of upregulated genes that may be closely related to HPCD-induced cell death, as well as their induction multipliers for each treatment condition, are shown in Table 4. According to the GO analysis, the major upregulated genes by HPCD treatment are associated with the pentose phosphate pathway (4 genes: *ZWF1*, *GND2*, *SOL3*, and *SOL4*); cell-wall organization or biogenesis (11 genes: *CRH1*, *GSC2*, *ECM8*, *HLR1*, *LDS2*, *SDP1*, *YMR084W*, *YMR085W*, *YNL194C*, *YGP1*, and *LST*); glutathione metabolism (2 genes: *GPX1* and *GSH*); protein refolding (6 genes: *HSP104*, *SSA1*, *SSA4*, *HSP82*, *SSE2*, and *CPR6*); phosphatidylcholine biosynthesis (1 gene: *OPI3*); and AdoMet synthesis (4 genes: *MET5*, *MET10*, *SAM1*, and *SAM2*). 

Most of the commonly downregulated genes in each treatment are related to the cell cycle, protein complex biogenesis, ribosome biogenesis, and RNA processing and metabolism. 

### 3.3. RT-qPCR Validation

Among the terms identified by the GO analysis, the RT-qPCR analysis was performed for five categories considered particularly relevant to HPCD-induced cell death. The induction multipliers identified in the RNA-seq results for *ZWF1*, *GPX1*, *OPI3*, *HSP104*, *SSA1*, *SSA4*, and *CRH1* were 1.49, 8.36, 2.30, 14.1, 6.19, 83.2, and 1.50, respectively (Table 4), while the mean RT-qPCR quantification values for *ZWF1*, *GPX1*, *OPI3*, *HSP104*, *SSA1*, *SSA4*, and *CRH1* were 1.30, 3.20, 1.80, 15.3, 7.35, 68.5, and 1.64, respectively (Figure 2). This suggests that the RNA-sequencing results and GO analysis are reliable.

### 3.4. Spot Assay for Phenotypic Analysis

Spot assays were performed on the deletion strains and the wild-type strain (*BY4741)* after 12 h of HPCD treatment (Figure 3). The dilution ratio is plotted on the X axis, while the Y axis displays the treatment condition. Figure 3 shows wild-type-strain lawns of before treatment formed on plates without dilution, 10-times dilution, and 100-times dilution. The colony-forming units (CFUs) before treatment were counted on the plate of 1000-times dilution. After 12 h of HPCD treatment, the CFUs could be easily counted on the no-dilution plate. Furthermore, only one colony was observed on the plate with the 100-times dilution, indicating that the wild-type-strain cells decreased greatly after 12 h of HPCD treatment. The *Δopi3* deletion strain formed a lawn on the plate without dilution after 12 h of HPCD treatment. The *Δopi3* strain and wild-type strain displayed no visible differences in appearance before treatment, indicating that the *Δopi3* strain cell number was higher than that of the wild-type strain after 12 h of HPCD treatment. However, only a few CFUs were observed on the plates containing the strains of *Δcrh1*, *Δgas1*, and *Δkre6* (no dilution), indicating that, compared to the wild-type strain, fewer cells were present after 12 h of HPCD treatment. In contrast, no substantial differences were observed on the *Δzwf1*, *Δgpx1*, *Δhsp104, Δgsh1*, and *Δchs5* plates compared to the wild-type-strain plate. These results were observed in the groups that received no treatment and those that were exposed to 12 h of HPCD treatment.

In conclusion, compared with the wild-type strain, the number of viable cells increased in *Δopi3* and decreased in *Δcrh1*, *Δgas1*, and *Δkre6*. No change was observed in the *Δzwf1*, *Δgpx1*, *Δhsp104, Δgsh1*, and *Δchs5* strains. 

### 3.5. Evaluation of the Effect of the Cell Wall on HPCD Susceptibility Using Protoplasts

Although it is axiomatic that the protoplasts are more sensitive to environmental stress than the vegetative cells, direct scientific evidence proving that is necessary. The effect of the cell wall on HPCD susceptibility was investigated using protoplasts. The viability of normal cells and protoplast cells is displayed in Figure 4. The viability was significantly reduced in protoplast cells. Microscopic analysis indicated that the normal cells maintained their shape even after 2 h of treatment, whereas the protoplast cells appeared to burst (Figure 4). This suggests that the cell wall is essential in protecting cells from damage by HPCD treatment.

## 4. Discussion

### 4.1. HPCD Treatment Increases Cell Requirements for NADPH

We observed that *ZWF1* in the cells treated for 1 h and 2 h was downregulated, while *ZWF1* in the cells treated for 4 h and *GND2*, *SOL3*, and *SOL4* in all of the treatment periods were upregulated. These genes are all related to the pentose phosphate pathway oxidative phase (Figure 5a), which is an important pathway for the generation of NADPH, a cofactor used in anabolic reactions [21,22]. The upregulation of the genes related to the pentose phosphate pathway no-oxidative phase was not detected, indicating that the cells have no requirement for higher levels of ribose 5-phosphate, which is the final product of the pentose phosphate pathway. Therefore, we suggest that the upregulation of *ZWF1*, *GND2*, *SOL3*, and *SOL4* resulted from the increased need for NADPH under HPCD-treatment conditions. The differences in susceptibility to HPCD between the mutant strains *Δzwf1* and wild strains were not identified (Figure 3). This can be attributed to the limited effect that *ZWF1* deletion has on the NADPH/NADP^+^ ratio due to the various alternate routes for generating the NADPH known to occur in *S. cerevisiae* [23,24]. Therefore, it is suggested that the increased requirement for NADPH induced by HPCD treatment was not the cause of cell death.

### 4.2. Cell-Wall Damage by HPCD Treatment and Its Effect on Cell Death

We observed that *CRH1* was not upregulated after 2 h of HPCD treatment; however, after a longer period of 4 h, *CRH1* was upregulated to 1.5 times the rate of normal untreated cells (Table 4). *CRH1* is responsible for the cross-linking between chitin and glucan in the cell wall [25]. These cross-links are essential for controlling yeast morphogenesis at the mother-bud neck [26], and accurate nuclear division is closely related to the formation of the mother-bud neck [27]. Therefore, we suggested that cell division was reduced by HPCD treatment under sub-lethal conditions, which led to the downregulation of *CRH1* in the 2 h treatment period (Table 4). We demonstrated that, as the time under HPCD conditions increased, the cells began to shift their self-protection mechanism to adapt to the stress, resulting in the upregulation of *CRH1* (Table 4). The upregulation of *CRH1* also indicates that the cell-wall structure was damaged by HPCD treatment. Furthermore, the *Δcrh1* mutant strain was more sensitive to HPCD than the wild-type strain (Figure 3). The result of RT-qPCR experiments (Figure 2) suggests that the RNA-sequencing results and GO analysis of *CRH1* were reliable.

To adapt to the HPCD stress, increased amounts of 1,3-β-D-glucan are required, as it is the most abundant component of the yeast cell wall and serves as the backbone of the structure [28]. The *GSC2* gene is responsible for producing 1,3-β-D-glucan synthase, which can convert one uridine diphosphate (UDP) glucose to 1,3-β-D-glucan (UDP-α-D-glucose + 1,3-β-D-glucan (n)→1,3-β-D-glucan (n+1) + UDP) [29]. The upregulated *GSC2* gene indicates that HPCD-treated cells may produce more 1,3-β-D-glucan (Table 4). 

The products of the *HLR1* gene are also considered to affect the cell-wall composition. Overexpression of *HLR1* partially suppressed the cell osmosensitivity [30]; therefore, the upregulation of *HLR1* resulted in the decreased osmosensitivity of HPCD-treated cells. Thus, we can assume that the upregulation of *GSC2* and *HLR1* is induced by the cell to strengthen the cell wall under HPCD-stress conditions. Figure 4 illustrates the sensitivity of protoplast cells to HPCD treatment. We found no surviving protoplast cells after 4 h of HPCD treatment. These results indicate that the cell wall is essential in protecting cells under HPCD stress. However, no differences in the HPCD inactivation effect were observed between Gram-positive bacteria and Gram-negative bacteria [1], thus suggesting that cell-wall disruption is not the key process resulting in lethal cell death.

We selected four deletion mutant strains for the *CRH1*, *CHS5*, *GAS1*, and *KRE6*, which are all related to cell-wall organization or biosynthesis, for phenotypic analysis. Those specific gene-deletion strains are known to be sensitive to sodium dodecyl sulfate (SDS) [31]. The *Δcrh1*, *Δgas1*, and *Δkre6* strains demonstrated an increased sensitivity to HPCD, while the *Δchs5* strain demonstrated the same level of sensitivity as the wild-type strain. The deletion of *CRH1* has been shown to affect the cross-linking of chitin to β (1–6) glucan [32], possibly weakening the cell-wall structure. The *GAS1* deletion strain had the highest sensitivity to HPCD and was found to have a lower β-glucan content in the cell wall, suggesting that the cell-wall structure could be weakened [33]. It is known that *Δkre6* strains have reduced amounts of alkali-insoluble glucans [34]. Since the alkaline solubility of β-d-glucans is a determinant of their three-dimensional structure [35], *KRE6* deletion may contribute to an alteration of the physical structure of the cell wall, leading to reduced resistance to HPCD treatment. *CHS5*, a gene related to chitin biosynthesis, may have an effect on recovery from cell-wall damage; however, it has a limited effect on the cell-wall structure. In our phenotypic analyses, the deletion of genes associated with cell-wall organization increased sensitivity to HPCD, whereas strains with knockouts of the genes involved in biosynthesis were comparable to wild-type strains. Hence, we suggest that the cell wall can protect the cell from HPCD damage; however, it is not the main cause of cell death. *SDP1* is a stress-inducible dual-specificity MAP (mitogen-activated protein) kinase phosphatase (Table 4). It can negatively regulate Slt2p MAP kinase via direct dephosphorylation [36]. MAP kinase pathways control cell growth, morphogenesis, proliferation, and stress responses [37]. Therefore, MAP kinase pathways are suppressed by the upregulation of *SDP1*, and cell growth, morphogenesis, proliferation, and stress responses are also influenced.

### 4.3. Broken Cell Organelle Membrane Is the Key Cause of Yeast Inactivation under Sublethal HPCD Condition

The function of the *YNL194C* gene is similar to that of the *SUR7* gene, which is responsible for sporulation and sphingolipid metabolism and subsequently affects the sphingolipid composition of the plasma membrane [38]. *YNL194* expresses a certain type of integral membrane protein [39]. As one of two main membrane protein groups, integral membrane proteins are tightly bound to the membranes by hydrophobic forces and can only dissociate from the membranes after treatments with surfactants (detergents) [40]. As a defense mechanism, *YNL194C* was upregulated in the HPCD-treated cells (Table 4), indicating that the activation or structure of YNL194Cp in the membrane was altered. Furthermore, another work proved that HPCD is highly similar to sodium dodecyl sulfate, a surfactant that helps reduce the surface tension of water [41]. Therefore, alteration of the YNL194Cp structure or activation thereof can be induced by high-pressure CO_2_, which may act as a surfactant. Hence, the alteration of the YNL194Cp structure or activation of the protein was considered the key factor resulting in cell death under sub-lethal HPCD conditions.

It is necessary to clarify which type of membrane damage (i.e., cell-membrane or organelle-membrane damage) was the main cause of the cell death resulting from HPCD treatment. Using a transmission electron microscope, it was proven that HPCD-inactivated cells may still be integral [42], meaning that the cellular membrane may maintain its integrity under HPCD stress conditions. Furthermore, according to our former work, no difference was found between the green fluorescent protein (GFP)-labeled cellular membrane in HPCD-treated and untreated cells, suggesting that the cellular membrane maintained its integrity after 4 h of HPCD treatment at 4.0 MPa and 30 °C. Contrastingly, the GFP-labeled protein delocalized on the organelle membrane, suggesting that the organelle membrane was damaged after treatment under the same conditions. These results indicate that damage to the major organelle membranes (such as the nuclear membrane and Golgi body membrane) was the main cause of cell death resulting from sub-lethal HPCD conditions (4.0 MPa and 30 °C for 4 h) [17]. Therefore, the alteration of the YNL194Cp structure or the activation thereof in the organelle membrane may be the key cause of cell death induced by sub-lethal HPCD conditions.

### 4.4. HPCD Treatment Possibly Induces Cell Nutrient Deprivation and Nitrogen Starvation

The expression of *YGP1* occurs when the cell is nutrient-deprived, especially when the glucose concentration in the medium falls below 1% and when nitrogen and phosphate are deficient [43]. The upregulation of *YGP1* indicates that the cell was suffering from nutrient deprivation. Interestingly, the nutrient level was sufficient for cell growth under our experimental conditions (YPD culture medium). Therefore, we presumed that the cell lost the ability to transfer nutrients from the medium to the cytoplasm. According to the analysis of the upregulated *YNL194C*, the alteration of the YNL194Cp structure or the activation induced the cell to take up less nutrients. Subsequently, the gene of *YGP1* was upregulated. 

We found that the *LST8* gene was significantly upregulated after HPCD treatment. *LST8* is required for amino acid permease Gap1p transport from the Golgi to the cell surface. *GAP1*, encoding a high-capacity permease that can transport all amino acids [44], can be regulated by the nitrogen level in the growth medium. *GAP1* transcription can be induced by nitrogen starvation [45]. As a component of a large complex on endosomal/Golgi membranes, LST8p is responsible for sensing intracellular nutrients and signaling metabolic pathways [46]. Hence, upregulation of *LST8* indicated that the cell was experiencing nitrogen starvation.

### 4.5. HPCD Treatment Possibly Induces Cell DNA Damage

*GSH1* encodes gamma-glutamylcysteine synthetase, which is associated with glutathione biosynthesis (response to intracellular oxidative stress) [47]. *GPX1* encodes glutathione peroxidase, which is induced by glucose starvation. The glucose-deficient condition was induced by the upregulated pentose phosphate pathway’s oxidative-phase-related genes, which consumed high levels of glucose (Table 4). Glutathione peroxidase converts glutathione to glutathione disulfide by reducing peroxides [48]. Therefore, *GSH1* and *GPX1* are upregulated to eliminate intracellular peroxides and protect DNA from suffering oxidative damage. Upregulation of *GSH1* and *GPX1* is also indicative of cell DNA damage. Furthermore, the *Δgsh1* mutant is prone to chromosomal damage [49]. Although the *Δgpx1* and *Δgsh1* mutants lost the ability to protect DNA from suffering oxidative damage, both *Δgpx1* and *Δgsh1* did not demonstrate increased sensitivity to HPCD treatment, according to the spot assay (Figure 3). Therefore, we can assume that the damage to cell DNA caused by HPCD treatment is not the cause of lethal cell death. A GO analysis and RT-qPCR experiments were performed on the *GPX1* gene after 4 h of HPCD treatment to verify the accuracy of the RNA-sequencing results. 

### 4.6. HPCD Treatment Induces Protein Denaturation and Aggregation

Six genes related to protein refolding were induced at high levels among all conditions. *HSP104* encodes a heat shock protein that cooperates with Ydj1p and Ssa1p to refold and reactivate previously denatured and aggregated proteins [50]. Hence, upregulated *HSP104* and *SSA1* indicate that some proteins may have aggregated or denatured due to HPCD exposure. Similarly, the activity of Na^+^/K^+^-ATPase, which is the main enzyme that maintains the balance of substances inside and outside of cells, was significantly reduced by HPCD treatment [51]. We also observed upregulation of *CPR6* that encodes peptidyl-prolyl cis–trans isomerase (cyclophilin), an enzyme that interconverts the cis and trans isomers of peptide bonds [52]. *CPR6* can also bind to HSP82p and contributes to chaperone activity. Increased CPR6p abundance occurs in response to DNA-replication stress [53]. Therefore, HPCD treatment results in damage to DNA replication.

Following the GO analysis, RT-qPCR experiments were performed for the *HSP104*, *SSA1*, and *SSA4* genes after the 4 h of HPCD treatment to verify the accuracy of the RNA-sequencing results. As displayed in Figure 2, the RT-qPCR result indicated that *HSP104* was upregulated by 15 times compared to the control sample. The RNA-seq analysis provided similar results; that is, HPCD treatment induced upregulation of *HSP104* by 14.1 times (Table 4). *SSA1* was upregulated by seven times the rate that was observed in the control sample according to the RT-qPCR result. RNA-seq results were similar and indicated an upregulation of 6.19 times the normal rate. *SSA4* was upregulated by 70 times according to the RT-qPCR experiment, whereas an upregulation by 83.2 times the normal rate was determined according to the RNA sequencing. The RNA-sequencing results for *HSP104*, *SSA1*, and *SSA4* were all in the error range of the RT-qPCR results; therefore, we can assume that the RT-qPCR results are in accordance with the RNA-sequencing results, indicating that the RNA-sequencing results and GO analysis are reliable. 

### 4.7. HPCD Treatment Possibly Increases the Cell Requirement for Phosphatidylcholine

*OPI3* is a key gene associated with the phosphatidylethanolamine methylation pathway of phosphatidylcholine biosynthesis in yeast. HPCD treatment can lead to a decreased ratio of phosphoglyceride to phosphatidylethanolamine, resulting in decreased cell-membrane stability [54]. We also evaluated the susceptibility of the *OPI3*-null strain to HPCD in order to explore the possibility that the *Δopi3* mutant strain is more sensitive to HPCD treatment than the wild strain. The results indicated that the *Δopi3* strain was less susceptible to HPCD when compared to the wild-type strain (Figure 3). The absence of the *OPI3* gene may result in a change in the phospholipid composition that makes up the cell membrane [55]. Therefore, the resistance of the *Δopi3* strain to HPCD was acquired due to the change in the structure of the cell membrane. 

Following GO analysis, RT-qPCR experiments were performed for the *OPI3* gene after 4 h of HPCD treatment to verify the accuracy of the RNA-sequencing results. As illustrated in Figure 2, *OPI3* was upregulated approximately two times compared to the control sample, in accordance with the RNA-seq results that indicated an upregulation by 2.30 times (Table 4). Therefore, the RNA-sequencing results and GO analysis for *OPI3* are reliable.

### 4.8. HPCD Treatment Increases the Cell Requirements for Hydrogen Sulfide and AdoMet Transfer in Response to Cell Nutrient Deprivation

Four genes, namely *MET5*, *MET10*, *SAM1*, and *SAM2*, which are related to AdoMet synthesis, were upregulated. *MET10* and *MET5* are responsible for transferring sulfite to hydrogen sulfide [56,57]. Subsequently, hydrogen sulfide will participate in the biosynthesis of various amino acids [58]. *SAM1* and *SAM2* are responsible for transferring L-methionine to S-adenosyl-L-methionine, which participate in spermine biosynthesis, spermidine biosynthesis, and S-adenosyl-L-methionine cycles I and II [59]. We believe that *MET10*, *MET5*, *SAM1*, and *SAM2* were upregulated in response to nutrient deprivation induced by HPCD to produce more cysteine (Figure 5d).

### 4.9. Possible Cell Lethal Mechanisms of HPCD on a Metabolic Level

According to our results, we suggest that inactivation or structure alteration of YNL194Cp in the organelle membrane may be the critical reason for HPCD-induced inactivation of cells under sub-lethal conditions (4.0 MPa and 30 °C; see Figure 5c and Figure 6). 

When HPCD is applied to cells, it acts as a surfactant that dissociates YNL194Cp (a type of integral membrane protein) from the organelle membranes. This results in a change in the structure of the organelle membranes. The stability of the membrane is reduced, and the permeability and fluidity are increased. As a self-protection mechanism, the cell has to upregulate the genes associated with the biosynthesis of membrane components to repair the damaged membrane. Therefore, *OPI3*, the key gene controlling phosphatidylcholine biosynthesis, is upregulated.

At the same time, due to the dissociation of YNL194Cp from the membrane, the genes related to protein folding or refolding are also upregulated. *HSP104* and *SSA1*, which are responsible for refolding and reactivating previously denatured or aggregated proteins, are all upregulated. These genes are also responsible for reactivating other denatured proteins or enzymes.

The upregulation of *CPR6* indicated that the DNA was suffering from replication stress. Furthermore, the upregulation of *GSH1* and *GPX1* indicated that the cell was attempting to protect its DNA from suffering oxidative damage.

Due to increased permeability and fluidity of the organelle membrane, the ability to transfer nutrients decreased due to the upregulation of *YGP1*. This resulted in nutrient deprivation, especially with regard to nitrogen. This was understandable; because certain proteins were denatured, the amino acid requirement would be enhanced for the cell to maintain various physiological activities.

We suggest that the intracellular metabolic pathways are influenced by HPCD due to the lack of nutrients. In order for the cell to acquire more nutrients, certain metabolic pathways, such as the pentose phosphate pathway (oxidative phase), glutathione metabolism, and AdoMet synthesis, were enhanced by several crucial enzymes, which are encoded by *MET10*, *MET5*, *SAM1*, *SAM2*, *GSH1*, *GPX1*, *ZWF1*, *SOL3*, *SOL4*, and *GND2*. In addition, *LST8* was also upregulated, contributing to GAP1 transferring more amino acids from the Golgi body to the cell surface.

We believe that there are two reasons for the upregulation of genes associated with cell wall biosynthesis. First, the cell wall was damaged by HPCD treatment; therefore, as a self-protection mechanism, the cell needed to produce more cell-wall constituents to maintain its integrity. Second, since the cell membrane (the second protective barrier) was damaged, the cell needed to strengthen its first protective barrier to protect itself. Hence, the genes related to the cell wall constituent’s biosynthesis (*GSC2*, *HLR1*, and *CRH1*) were upregulated.

Due to altered membrane constituents, the cell could not optimally transfer nutrients. As a result, cell growth and division were suppressed. The upregulation of *SDP1* would occur and result in MAP kinases inactivation, which is responsible for cell growth, morphogenesis, proliferation, and stress responses. Therefore, the cell would be inactivated.

## 5. Conclusions

In this work, six metabolic pathways were identified among the induced genes of the cells that experienced sub-lethal HPCD conditions (4.0 MPa, 30 °C, and 1–4 h). Based on the upregulated *ZWF1*, *GND2*, *SOL3*, and *SOL4* genes, it was concluded that the NADPH requirement of HPCD-injured cells increased. Upregulation of *CRH1* and the increased sensitivity of *Δcrh1* mutant strain to HPCD indicated that cell division could be suppressed by HPCD treatment. The increased sensitivity of protoplasts to HPCD also suggested that the cell wall is essential in protecting cells from damage by HPCD treatment. Furthermore, this work demonstrated that the alteration of the YNL194Cp structure or activation thereof might be the key factor associated with HPCD-induced cell death. However, further research is required to substantiate our claim. We also found that upregulation of *YGP1* and *LST8* corresponded with nutrient deprivation. Furthermore, we demonstrated that DNA damage was not the cause of cell death, as both *Δgpx1* and *Δgsh1* mutants (lost the ability to protect cell DNA from suffering oxidative damage) did not present an increased sensitivity to HPCD treatment. Finally, we discovered that *MET10*, *MET5*, *SAM1*, and *SAM2* were upregulated in response to HPCD-induced nutrient deprivation in an attempt to produce more cysteine.

Strains with gene deletions related to damage recovery and protein refolding did not present a significant level of susceptibility, therefore suggesting that the lethal strength of HPCD may drastically affect yeast cells and make recovery difficult. In addition, the *Δopi3* strain, in which the composition of the cell membrane was altered, became less sensitive, suggesting that yeast may be able to adapt to HPCD by replacing components of the cellular membrane. In strains deficient in genes related to cell-wall organization and biosynthesis, an altered cell-wall structure was observed, and these strains became vulnerable to HPCD. These phenotypic analyses suggest that yeast cannot recover from HPCD treatment under lethal conditions and that resistance to HPCD is due to changes in physical structures, such as the cell wall and cell membrane. Therefore, HPCD may be superior because fewer cells are likely to recover after treatment.

Although we demonstrated the possible mechanisms responsible for cell death caused by HPCD on a metabolic level, our results were based solely on the analysis of RNA sequencing and gene ontology analysis, which do have certain limitations. Therefore, further research is necessary to provide direct evidence on the mechanisms responsible for HPCD-induced cell death. Despite that, our study may still have significant implications for developing HPCD technology within the food industry.

## Figures and Tables

**Figure 1 jof-08-01011-f001:**
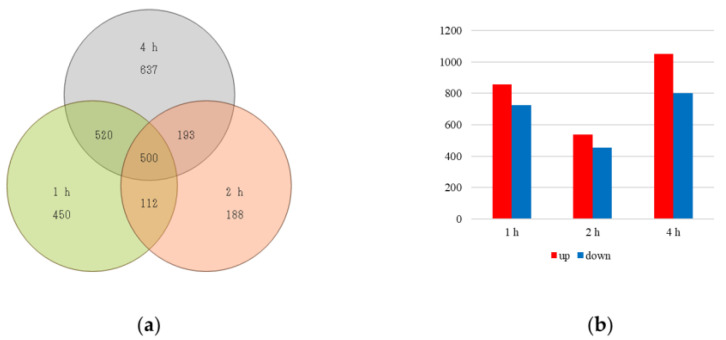
(**a**) Venn diagrams of DEGs in the three groups marked as 1 h treatment versus before treatment, 2 h treatment versus before treatment, and 4 h treatment versus before treatment, respectively. (**b**) Up- or downregulated DEGs in pairwise comparisons marked as 1 h treatment versus before treatment, 2 h treatment versus before treatment, and 4 h treatment versus before treatment, respectively.

**Figure 2 jof-08-01011-f002:**
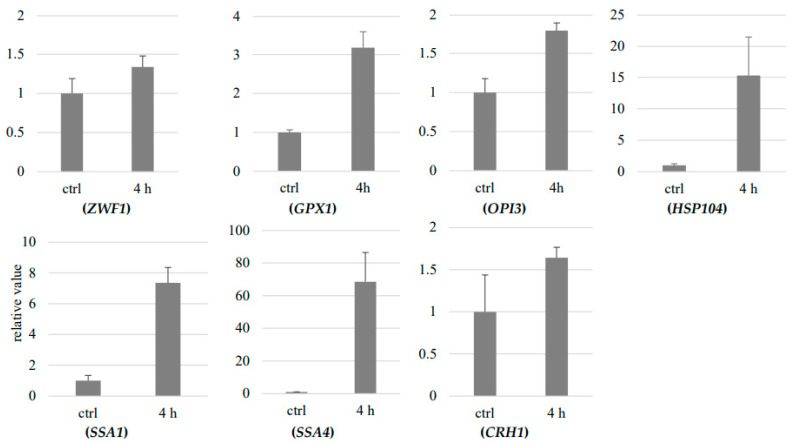
RT-qPCR validation. X axial shows the samples which are treated by HPCD 4.0 MPa 30 °C 4 h. Y axial shows relative value. The word ‘ctrl’ means ‘control sample’.

**Figure 3 jof-08-01011-f003:**
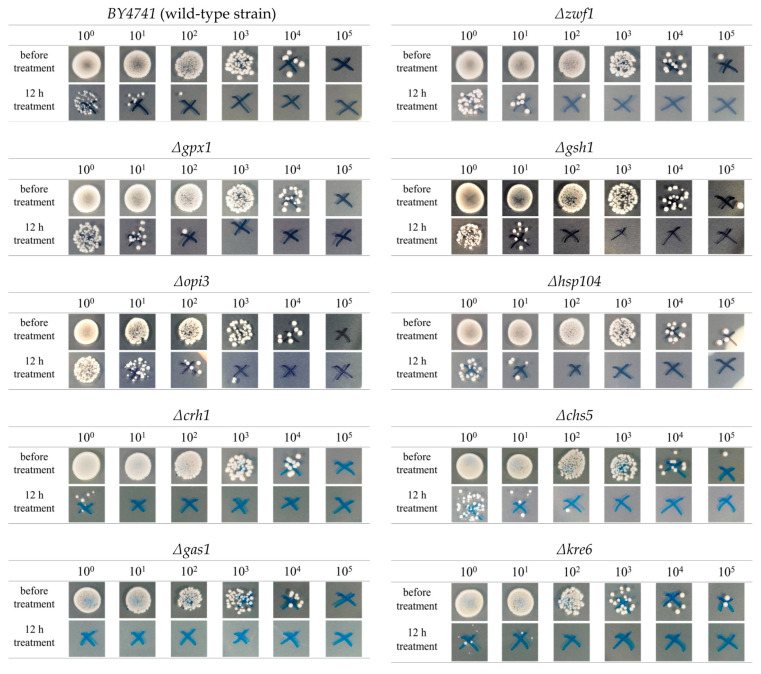
Different strains’ spot assays before treatment and after 12 h HPCD treatment (4.0 MPa, 30 °C). X axial shows the dilution ratio. Y axial shows treatment condition. Different strain names are presented on the top of each group of the spots’ results.

**Figure 4 jof-08-01011-f004:**
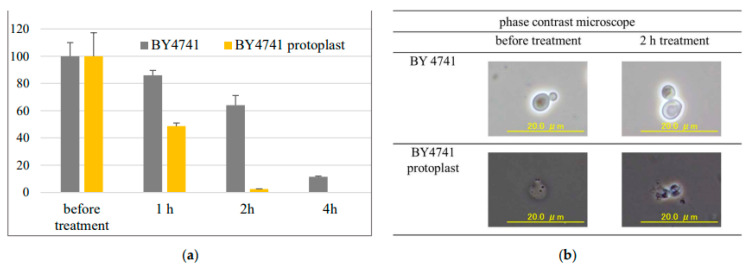
The sensitivity of normal cells and protoplast cells to HPCD treatment. (**a**) Survival of yeast cell rates after HPCD treatment. The percentage before treatment was set to 100%. (**b**) Visualization of yeast cells by phase-contrast microscopy.

**Figure 5 jof-08-01011-f005:**
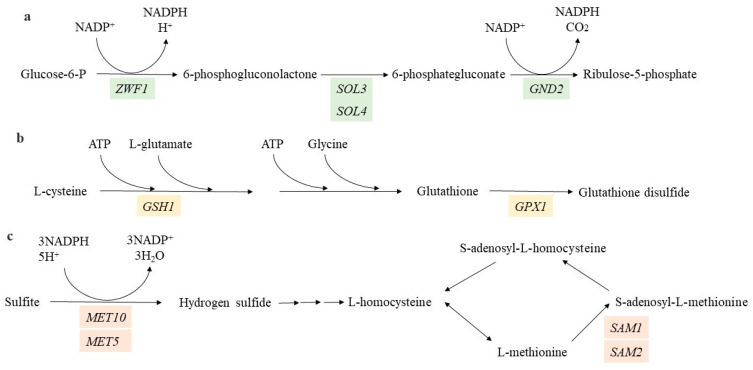
(**a**) Pentose phosphate pathway (oxidative phase), (**b**) glutathione metabolism, (**c**) AdoMet synthesis, and (**d**) a combination of ABC. The green color indicates the genes that are related to the oxidative phase in the pentose phosphate pathway, the red color indicates the genes that are related to AdoMet synthesis, and the yellow color indicates the genes that are related to glutathione metabolism.

**Figure 6 jof-08-01011-f006:**
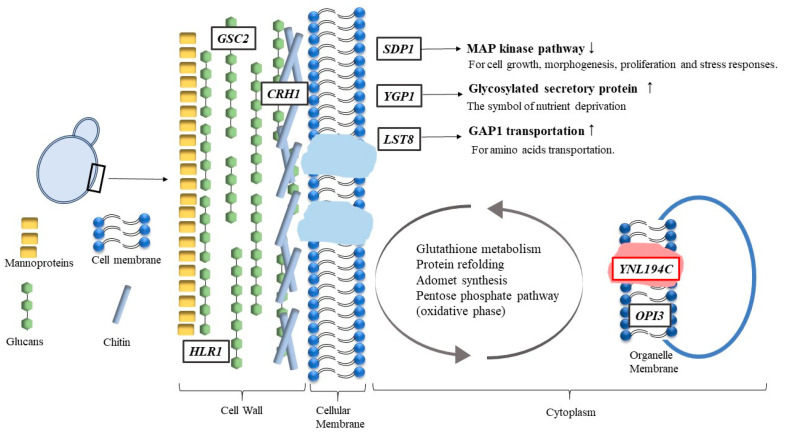
The sketch of the possible cell lethal mechanisms caused by HPCD on the metabolic level.

**Table 1 jof-08-01011-t001:** Yeast strains used in this study.

Strain	Genotype
*S. cerevisiae BY4741*	MATa his3Δ1 leu2 Δ0 met15Δ0 ura3Δ0
*S. cerevisiae BY4741 Δzwf1*	MATa his3Δ1 leu2 Δ0 met15Δ0 ura3Δ0 YNL241C::KanMX
*S. cerevisiae BY4741 Δopi3*	MATa his3Δ1 leu2 Δ0 met15Δ0 ura3Δ0 YJR073C::KanMX
*S. cerevisiae BY4741 Δgpx1*	MATa his3Δ1 leu2Δ0 met15Δ0 ura3Δ0 YKL026C::KanMX
*S. cerevisiae BY4741 Δgsh1*	MATa his3Δ1 leu2 Δ0 met15Δ0 ura3Δ0 YJL101C::KanMX
*S. cerevisiae BY4741 Δhsp104*	MATa his3Δ1 leu2Δ0 met15Δ0 ura3Δ0 YLL026W::KanMX
*S. cerevisiae BY4741 Δcrh1*	MATa his3Δ1 leu2 Δ0 met15Δ0 ura3Δ0 YGR189C::KanMX
*S. cerevisiae BY4741 Δchs5*	MATa his3Δ1 leu2 Δ0 met15Δ0 ura3Δ0 YLR330W::KanMX
*S. cerevisiae BY4741 Δgas1*	MATa his3Δ1 leu2 Δ0 met15Δ0 ura3Δ0 YMR307W::KanMX
*S. cerevisiae BY4741 Δkre6*	MATa his3Δ1 leu2 Δ0 met15Δ0 ura3Δ0 YPR159W::KanMX

**Table 2 jof-08-01011-t002:** Primers used in this study.

Gene	Forward Primer	Reverse Primer	Product Size
*ACT1*	ACATCGTTATGTCCGGTGGT	CCACCAATCCAGACGGAGTA	142 bp
*HSP104*	TGCTACCGCCGCTGATTTAA	GGATCATGGAGTTGGCACCA	116 bp
*SSA1*	TCTCCTCCATGGTCTTGGGT	AACCAGCAATGGTACCAGCA	144 bp
*SSA4*	ATTGCGTATGGGCTGGACAA	GTGTCACCAGCAGTAGCCTT	137 bp
*GPX1*	GGCAAAAGCAAGATCCCGTC	ACCACCTTCCCATTTCGGTC	118 bp
*CRH1*	GGCTGCCGAAAGTACTGCTA	CCGGCGTACAACCTGTAGTT	92 bp
*ZWF1*	TCGCATCGGGTGTCTTCAAA	AGCATTTGACAGACCAGGGG	113 bp
*OPI3*	CATTGCGTGAACAGCCTACG	GTCACCCAAGTACGTCCCTG	140 bp

**Table 3 jof-08-01011-t003:** Quality statistics of clean sequencing data. Q20 and Q30 refer to the proportion of base calls with Phred scores > 20 or 30 in the total bases. Higher scores ensured the base-calling accuracy and data quality. The sample before treatment has two biological replicates. The c_1 and c_2 are before HPCD treatment; t_1h is after 1 h HPCD treatment; t_2h is after 2 h of HPCD treatment; and t_4h is after 4 h HPCD treatment.

Sample	Raw Reads	Clean Reads	Raw Base (G)	Clean Base (G)	Q20 (%)	Q30 (%)	Mapping Rate (%)
c_1	16555668	16104561	5.0	4.8	97.84	93.63	97.63
c_2	20000002	19561112	6.0	5.9	97.81	93.53	97.84
t_1h	15087780	14741627	4.5	4.4	97.86	93.65	97.37
t_2h	12503099	11982433	3.8	3.6	97.94	93.88	97.36
t_4h	17587792	17204215	5.3	5.2	97.85	93.63	97.55

**Table 4 jof-08-01011-t004:** Major categories were identified by GO analysis and their induction multipliers for each treatment condition.

Gene Name	1 h	2 h	4 h	Description
Pentose phosphate pathway, oxidative phase				
*ZWF1*	0.86	0.68	1.49	Glucose-6-phosphate dehydrogenase (G6PD)
*GND2*	2.03	4.22	7.82	6-phosphogluconate dehydrogenase (decarboxylating)
*SOL3*	3.83	2.95	2.34	6-phosphogluconolactonase
*SOL4*	5.79	2.08	6.22	6-phosphogluconolactonase
Cell wall organization or biogenesis				
*CRH1*	0.64	0.80	1.50	Chitin transglycosylase
*GSC2*	9.59	5.84	13.8	Catalytic subunit of 1,3-beta-glucan synthase alternate catalytic subunit
*ECM8*	2.19	2.28	4.12	Non-essential protein of unknown function
*HLR1*	2.04	2.21	2.95	Protein involved in regulation of cell wall composition and integrity and response to osmotic stress
*LDS2*	5.84	2.14	8.82	Protein of unknown function
*SDP1*	2.26	3.31	6.27	Stress-inducible dual-specificity MAP kinase phosphatase
*YMR084W*	7.11	4.47	11.0	Putative protein of unknown function
*YMR085W*	5.38	3.59	7.83	Putative protein of unknown function
*YNL194C*	7.64	4.22	14.1	Integral membrane protein required for sporulation and plasma membrane sphingolipid content
*YGP1*	2.27	2.02	3.19	Cell-wall-related secretory glycoprotein
*LST8*	10.62	4.77	35.4	Protein required for the transport of amino acid permease Gap1p from the Golgi to the cell surface; component of the TOR signaling pathway
Glutathione metabolism				
*GPX1*	7.48	4.62	8.36	Phospholipid hydroperoxide glutathione peroxidase
*GSH1*	3.01	2.28	3.29	Gamma glutamylcysteine synthetase catalyzes the first step in glutathione (*GSH*) biosynthesis
Protein refolding				
*HSP104*	9.62	1.93	14.1	Heat shock protein that cooperates with Ydj1p (*Hsp40*) and Ssa1p (*Hsp70*) to refold and reactivate previously denatured, aggregated proteins
*SSA1*	4.59	2.08	6.19	ATPase involved in protein folding and nuclear localization signal (NLS)-directed nuclear transport
*SSA4*	61.9	9.64	83.2	Heat shock protein that is highly induced upon stress
*HSP82*	11.4	2.33	17.6	*Hsp90* chaperone required for pheromone signaling and negative regulation of Hsf1p
*SSE2*	8.30	2.13	10.4	Member of the heat shock protein 70 (*HSP70*) family
*CPR6*	5.82	2.26	9.13	Peptidyl-prolyl cis–trans isomerase (cyclophilin)
Phosphatidylcholine biosynthesis				
*OPI3*	2.35	2.64	2.30	Phospholipid methyltransferase (methylene-fatty-acyl-phospholipid synthase)
AdoMet synthesis				
*Met5*	1.77	2.43	2.04	Sulfite reductase beta subunit, involved in amino acid biosynthesis
*Met10*	2.24	2.31	2.26	Subunit alpha of assimilatory sulfite reductase
*SAM1*	4.87	2.83	3.20	S-adenosylmethionine synthetase
*SAM2*	8.80	8.23	6.13	S-adenosylmethionine synthetase

## Data Availability

Not applicable.

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
