# Peer review of "Transcriptome Analysis of the Influence of High-Pressure Carbon Dioxide on Saccharomyces cerevisiae under Sub-Lethal Condition"

_jof, 2022, doi:10.3390/jof8101011_

Round 1
Reviewer 1 Report
In this manuscript, Takashi et al. explored the molecular mechanism of cell damage by sub-lethal high-pressure carbon dioxide (HPDC). The evidence provided by the authors is sufficient to suggest that the deleterious effect of HPDC on cell viability occurs via cell-wall damage in Saccharomyces cerevisiae. Authors carefully discuss and associate its transcriptomic results with proposing a mechanism involving all the relevant genes differentially transcribed. Also, in the discussion section authors indicate the gaps in the study. It would be very tempting to ask the authors for additional experiments to test their main idea. However, the ideas exposed by the author could be explored in future investigations. Therefore, this manuscript is reliable to be published with some minor revisions.
Include in figure 2 the gene names as titles of the graphs to facilitate figure interpretation.
In figure 3, a direct comparison with WT could be more informative and simplify reader understanding.
Author Response
We appreciate reviewer's professional comments. A point-by-point response is presented as following.
Comment1: Include in figure 2 the gene names as titles of the graphs to facilitate figure interpretation.
Response1:The gene names have been already typed on the bottom of each figures separately (in Figure 2).
Comment2: In figure 3, a direct comparison with WT could be more informative and simplify reader understanding.
Response2: A direct comparison with wild-type strain has been written in part of [3.4 Spot Assay for Phenotypic Analysis.].
Comment3: English language and style are fine/minor spell check required.
Response3: The English editing of this manuscript has been corrected by Editage before the first submission. However, due to the time limitation, we seriously checked the English language and style by ourselves for revised manuscript.
Reviewer 2 Report
In this paper, the authors aim to understand the molecular mechanism of pasteurization by high-pressure carbon dioxide (HPCD). To do so, the authors applied sub-lethal HPCD to S. cerevisiae cells and studied its transcriptomic response. Based on the function of upregulated metabolic genes, the authors propose potential pathways affected by HPCD and propose that YNL194C is a key factor in mediating HPCD-induced cell death. While interesting, this model is entirely speculative and unsubstantiated because of several important shortcomings of the paper:
1) It seems that no replicates were made for RNA-seq, which results in the inability to estimate biological variation, FDR, and therefore cannot provide statistical significance. While it is good that RT-qPCR for specific genes agreed with RNA-seq data, this can only be tested for a subset of genes and cannot substitute for proper biological replicates.
2) The authors discuss a number of up-regulated DEGs (enumerated in sections 4.1 to 4.8), but down-regulated genes, which may also be essential in the HPCD response, are entirely ignored – and no table is given to illustrate specific genes in that category.
3) No further experiments were conducted to assay the role of YNL194C. Its deletion is viable (cf. SGD database) and it would therefore be a simple experiment to at least check if this mutant is hypersensitive to HPCD. In fact, none of the discussed points on potential mechanisms is substantiated by further experiments; this entire study therefore appears very preliminary and not yet at a publication-ready stage.
Other issues:
The start of the Results section highlights the Phred score for the read. While this is a good metric, it would be more appropriate to keep this QC step in the Material & Methods section. On the other hand, a much more important metric is not mentioned in Table 3 and in this section: the mapping rate. How many % of the reads mapped to the transcriptome? This measure can be added to Table 3 as well as following the Phred score paragraph (also preferably in Material & Methods: the Results section can start at the current line 189).
When DEGs are mentioned in line 189, it would be useful to mention what was the criteria for defining differential expression (fold-change, p-val?). A useful measure of this, rather than an absolute number of genes (as in Fig 1b, which will vary a lot depending on the criteria used), would be to generate a ‘volcano-plot’, which would also visualize the thresholds used. This could become Fig 1C. Another possibility could be a clustered heatmap comparing gene expression levels across all samples.
Table 3 mentions 5 samples. Does this indicate that each transcriptome condition corresponds to n=1? If so, this is a major weakness in the paper. Not only many statistical measures cannot be applied with n=1, but also it is known that transcriptome analysis is subject to technical and biological variation. The ENCODE guidelines from 2017 give a minimum n=2 for RNA-seq experiments, and it is common practice in several labs to make this n=3. Absolute numbers of DEGs only have meaning when knowing (1) the thresholds used and (2) the biological variability of the condition of interest.
line 211: how were log10 p-values calculated considering n=1? Were the treated libraries combined to obtain n=3?
line 223: upregulated genes involved in metabolism are shown in the text and in Table 4, but down-regulated genes, which may be as important in the HPCD process as up-regulated genes, are not mentioned and only referred to by categories and “data not shown”. Why is this the case? It would be justified to make another Table with these – or at the very least, include a Supplemental Table with all of this information. The presence of GO categories such as cell cycle, protein biogenesis and metabolism clearly indicate that these could play a key role in the process. Consider this example: one or several essential genes, when fully inhibited, would result in cell death.
Table 4 does not specify the units for columns 2,3 and 4. Are these fold changes or log fold-changes? If the latter, are these log2 or log10?
Figure 2 legend should specify if the error bars correspond to standard deviation or standard error.
The spot assay in Figure 3 is organized in an uncommon way, splitting every spot. How were plates organized for this experiment? If the spots were contiguous, then it may make more sense to split only horizontally?
The resolution of Figure 4 is too low in the current version of the manuscript; the legends and images could be increased in size to make it clearer.
Figure 5 should include legends explaining the colors for the mentioned genes.
While the discussion makes some interesting points, it reads like an enumeration of genes observed in the transcriptomic data, and apart from the spot assay from Figure 3 no experiments were made to test any of the speculations proposed in sections 4.1 to 4.8. In some cases, these experiments would be relatively straight-forward. Note also that the discussion lacks all down-regulated genes (see comment above).
line 364: “bactericidal” -> yeast is not a bacterium.
line 375-380: the authors mention the interesting observation that HPCD treatment may be similar to SDS treatment. A simple experiment to test this would be to repeat a spot assay with SDS plates on the wild-type and YNL194C deletion strain. The current results suggest that YNL194C deletion mutants would be hypersensitive to SDS.
line 402-406: the authors propose that HPCD results in a nutrient limitation-like condition for the cell, resulting in YGP1 upregulation. This is also a testable hypothesis – the authors can monitor the intake of a nutrient or analog.
section 4.5 (line 417) proposes that HPCD induced DNA damage. However, the upregulation of GSH1 and GPX1 is not by itself evidence of DNA damage. To do so, the authors would have to check mutation rates at marker sites (or by sequencing), or show direct evidence of DNA damage (for example, phospho-H2A.X on chromatin by Western blot).
Reviewer 3 Report
Accept but need minor correction
Minor comments were listed below.
Keep identical for the dash (– or -) and journal name format in References section. Should be state identical in text for 4MPa (or 4.0MPa).
The terms “yeast” and “yeasts” should be classified in text.
Line 86: correct ‘S.cerevisiae BY471 Δzwf1’ to ‘S. cerevisiae BY4741 Δzwf1’
line 157 what was the condition of HPCD? 4 MPa, 30 ℃?
line 495: MET10, MET5 should be italic, check other gene names in text.
Author Response
We deeply appreciate reviewer’s professional comments. Our point-to-point response is displayed as follows.
Comment 1: Keep identical for the dash (– or -) and journal name format in References section.
Response 1: The dash was kept as (-) across the references section and all the text.
Comment 2: Should be state identical in text for 4MPa (or 4.0MPa).
Comment 3: The terms “yeast” and “yeasts” should be classified in text.
Response 2 & 3: ‘4.0 MPa’ and ‘yeast’ were used across all the text.
Comment 4: Line 86: correct ‘S.cerevisiae BY471 Δzwf1’ to ‘S. cerevisiae BY4741 Δzwf1’
Response 4: In the section of [2.1 Stains and Culture Media], ‘S.cerevisiae BY471 Δzwf1’ has been corrected to ‘S. cerevisiae BY4741 Δzwf1’
Comment 5: line 157 what was the condition of HPCD? 4 MPa, 30 ℃?
Response 5: A condition of ‘(4.0 MPa, 30 °C)’ was added following ‘HPCD’ in the third line of [2.6 Spot Assay].
Comment 6: line 495: MET10, MET5 should be italic, check other gene names in text.
Response 6: MET10, MET5, and other gene names have been corrected by italic.
Comment 7: English language and style are fine/minor spell check required
Response 7: The English editing of this manuscript has been corrected by Editage before the first submission. However, due to the time limitation, we seriously checked the English language and style by ourselves for the revised manuscript.
Reviewer 4 Report
In this manuscript, Takahashi, Yu, and Ywahashi performed a timely study regarding the effect of high-pressure CO2 (HPCD) on Saccharomyces cerevisiae cells. The study is important as HPCD is a novel non-thermal pasteurization technology whose effect on cell death is largely not understood. To investigate the detailed lethal mechanism caused by HPCD treatment, the authors approached the topic by transcriptome analysis through RNA sequencing and RT-qPCR and by gene ontology analysis.
The manuscript is very well written, bringing robust evidence of the effect of sub-lethal HPCD on yeast transcriptome. There are minor issues that the authors need to address before the manuscript can be accepted for publication.
1) Line 158: please explain the meaning of „100 to 10-5”. Is it „100 to 10-5”?
2) Figure 2: please explain the meaning of „ctrl” in the figure legend.
3) Section 3.4 (lines 241-242) and Figure 3: use lower-case Latin letters when naming knockout mutants.
4) The rationale of the experiment using protoplasts is not clear (section 3.5. The fact that protoplasts would be less viable than the walled cells is rather axiomatic. Please elaborate more in this direction.
Author Response
We deeply appreciate reviewer’s professional comments. Our point-to-point response is displayed as follows.
Comment 1: Line 158: please explain the meaning of „100 to 10-5”. Is it „100to 10-5”?
Response 1: We have revised original contents in [2.6 Spot Assay] by ‘(100 to105 times dilution)’.
Comment 2: Figure 2: please explain the meaning of „ctrl” in the figure legend.
Response 2: We added a sentence of ‘The word ‘ctrl’ means ‘control sample’ in the Figure 2 legend.
Comment 3: Section 3.4 (lines 241-242) and Figure 3: use lower-case Latin letters when naming knockout mutants.
Response 3: All the knockout mutant names are revised by using lower-case Latin letters across the whole text.
Comment 4: The rationale of the experiment using protoplasts is not clear (section 3.5. The fact that protoplasts would be less viable than the walled cells is rather axiomatic. Please elaborate more in this direction.
Response 4: Although the protoplasts would be less viable than the walled cells is rather axiomatic, directly scientific evidence is still necessary for proving that the cell wall is essential in protecting cells from damage by HPCD treatment. The sentence ‘Although it is axiomatic that the protoplasts are more sensitive to environmental stress than vegetative cells, the directly scientific evidence of proving that is necessary. The effect of the cell wall on HPCD susceptibility using protoplasts was invested in this work.’ and ‘This suggests that the cell wall is essential in protecting cells from damage by HPCD treatment.’ are supplied in section of [3.5. Evaluation of the Effect of the Cell Wall on HPCD Susceptibility Using Protoplasts].
Comment 5: Moderate English changes required.
Response 5: The English editing of this manuscript has been corrected by Editage before the first submission. However, due to the time limitation, we seriously checked the English language and style by ourselves for the revised manuscript
Reviewer 5 Report
Comments on the manuscript “Transcriptome Analysis of the Influence of High-Pressure Carbon Dioxide on Saccharomyces cerevisiae under Sub-Lethal Condition” by Takahashi et al. (jof-1886112):
High pressure carbon dioxide (HPCD) treatment is a new promising mechanism for non-thermal pasteurization and may offer advantages against traditional methods in future application. However, the mechanism leading to lethality of organisms remains unclear. To obtain experimental evidence for possible mechanisms, Takahashi et al. studied the response of the eukaryotic model organism S. cerevisiae on sub-lethal HPCD conditions as a new stress factor. The authors used RNA seq as a method to identify yeast genes showing differential expression in stressed cells compared to non-stressed cells. Following a gene ontology analysis, they next validated differential expression of selected candidate genes by qRT-PCR. Finally, the authors comparatively investigated whether selected null mutants are less resistant against HPCD stress than wild-type cells. In summary, this is an interesting manuscript on candidate genes affected by a HPCD as a new stress mechanism which should be published once a number of weak points have been resolved.
Special comments:
- Results section, l. 180 ff: Results directly start with a presentation of quality data of the differential gene expression analysis. Although it is essentially clear from the previous sections what the authors wish to investigate, the results section nevertheless should start with a clear description of the performed experiments (information on strain used, growth and stress conditions, description of experimental procedures, etc). After having provided such information, the authors should continue with data on quality aspects of their sequencing work.
- Fig. 1 a: This figure does not allow distinguishing between genes being up- or down-regulated. Eventually, two Venn diagrams showing up- and down-regulated genes separately may be helpful.
- A major problem of this work is the choice of genes selected for a comparative analysis against HPCD stress by the spot assay. From my point of view, it would be plausible to select genes with strongest induction by HPCD for comparatively studying mutant phenotypes (as long as the mutant is viable). For several genes showing a strong induction by HPCD, no mutants were investigated. As an example, SSA4 showed the strongest up-regulation of all genes (Table 4; confirmed by qRT-PCR, Fig. 2) but a ssa4 mutant was not studied for its response against HPCD treatment. Since the four SSA genes may be partially redundant for protein refolding, it could be necessary to investigate a ssa1 ssa4 double mutant. The same argument is true for the beta-glucan synthase gene GSC2 which was strongly up-regulated by HPCD but a gsc2 mutant was not investigated. On the contrary, some genes such as GAS1 and KRE6 obviously are not influenced by HPCD (they are not listed in Table 4 as DEG) but gas1 and kre6 null mutants were studied for their phenotype against HPCD stress. The authors should give convincing reasons for their selection of mutants investigated by the spot assay. In conclusion, additional spot assays with mutants which correspond to DEG should be performed.
- Discussion, l. 273 ff, general comment: Some sub-sections of the discussion can be shortened as they simply repeat findings from the Results section (e. g. data from RNA seq and qRT-PCR).
- Discussion of cell wall damage by HPCD treatment: To me, it appears highly plausible that HPCD stress may cause cell wall damage. However, the authors´ comments on the importance of cell wall damage are not sufficiently clear and contradictory: “... cell wall disruption is not the key process resulting in lethal cell death” (l. 335-336); “... that the cell wall is essential in protecting cells from damage by HPCD treatment. However, we found that damage to the cell wall was not the main cause of lethal cell death” (l. 554-556). To avoid confusion, these comments must be modified and a more precise conclusion is required.
- The possible importance of YNL194C is strongly overestimated in the discussion and is based solely on differential gene expression: “Hence, the alteration of the YNL194Cp structure or activation of the protein was considered the key factor resulting in cell death under sub-lethal HPCD conditions” (l. 378-380). No experimental evidence from this manuscript supports this view. The authors should at least study the phenotype of a ynl194c null mutant by spot assay (according to SGD, a null mutant is viable).
- Certain headlines of the Discussion section are highly speculative and are not sufficiently supported by experimental evidence. This is true for sections 4.4. (HPCD treatment induces cell nutrient deprivation and nitrogen starvation, l. 396), 4.5. (HPCD treatment induces cell DNA damage, l. 417) and 4.7. (HPCD treatment increases the cell requirement for phosphatidylcholine, l. 468). I should mention a comment taken from the authors´ conclusions (l. 579-581): “Although we demonstrated the possible mechanisms responsible for cell death caused by HPCD on a metabolic level, our results were based solely on the analysis of RNA sequencing and gene ontology analysis, which do have certain limitations.” This is a very important comment and I suggest moving it to the beginning of the Discussion section. Consequently, the authors must be much more cautious with their speculations on possible mechanisms responsible for HPCD lethality. The headline of section 4.5. (“HPCD treatment induces cell DNA damage”) is not supported by experimental findings. Up-regulation of GSH1 and GPX1 indicates oxidative stress. Genes of the DNA damage response were obviously not up-regulated (e. g. RAD genes). The headline of section 4.7. (“HPCD treatment increases the cell requirement for phosphatidylcholine”) should be much more cautious. Comparative studies on PC concentration before and after HPCD treatment would be necessary to justify this conclusion.
- In general, the conclusions can be deleted because this section mostly repeats comments from the Discussion.
Minor comments:
- l. 26: “phosphatidylcholine bio-system”; presumably, the authors mean “phosphatidylcholine biosynthesis”.
- Recessive mutations in S. cerevisiae are generally indicated by lower case letters (e. g. delta zwf1 as in Table 1, not delta ZWF1 as in Fig. 3 and in section 3.4, also in the discussion).
- Fig. 4: For a better visualization, increase the size of Fig. 4 a, b. On the contrary, spots shown in Fig. 3 can be downsized.
Round 2
Reviewer 2 Report
In this revised version of the manuscript, the text has been adjusted, but unfortunately the authors have not addressed the raised issues, including for simple points not requiring experiments (such as giving the % mapping rate rather than the Phred scores - the % mapping rate is very informative, while the Phred score very little; and only reflects that the sequencing reaction worked irrespective of the template).
For example, a major issue is that of the number of replicates. The authors do not provide an accession number for the RNA-seq data and therefore neither the % mapping nor the experimental design can be verified. Mixing 3 RNA-seq libraries together with the same barcode is certainly not standard and not recommended for DEG analysis. Most labs would make 3 RNA-seq libraries with different barcodes, corresponding to 3 biological replicates giving independent FASTQ files, independent mapping rates and independent Phred scores. If this is what the authors indeed did, then the explanation should be clarified, and an accession number should be provided to the data on a database (SRA/GEO/etc). The author response of adding one sentence to the material & methods is insufficient.
While simply checking a YNL194c mutant response to HPCD or SDS is indeed insufficient for a full paper, it would here provide strong evidence for the claims made in this study, and given the simplicity of these experiments and how preliminary this study appears, it is hard to justify the authors' unwillingness to address some of these points.
Overall, it appears that the only addressed comments are point 10, 11 and 13, and that every other comment from the first review still needs to be addressed.
Reviewer 5 Report
Comments on the manuscript “Transcriptome Analysis of the Influence of High-Pressure Carbon Dioxide on Saccharomyces cerevisiae under Sub-Lethal Condition” by Yu et al. (jof-1886112):
In most cases, the authors responded adequately and introduced a number of alterations into the manuscript. I am still not really convinced with the choice of null mutants for HPCD spot assay but I accept the authors´ explanation. There is one minor problem which should be solved (comment 9, Abstreact, l.26): The authors confirm that “phosphatidylcholine biosynthesis” instead of “‘phosphatidylcholine biosystem” is correct but they did not change this in the revised manuscript. This should be done prior to acceptance.
Author Response
Thank to reviewer’s comment, we have correctly changed the word of biosynthesis instead of biosystem in the Abstract and [3.2 GO Enrichment Analysis of the DEGs] parts.